# The Oxygen Reduction Performance of Pt Supported on the Hybrid of Porous Carbon Nanofibers and Carbon Black

**DOI:** 10.3390/ma15134560

**Published:** 2022-06-29

**Authors:** Tongyu Zhou, Jingjing Zhang, Shenglin Yang, Junhong Jin, Biao Wang, Guang Li

**Affiliations:** State Key Laboratory for Modification of Chemical Fibers and Polymer Materials, College of Materials Science and Engineering, Donghua University, Shanghai 201620, China; zty_call@163.com (T.Z.); slyang@dhu.edu.cn (S.Y.); jhkin@dhu.edu.cn (J.J.); wbiao2000@dhu.edu.cn (B.W.)

**Keywords:** proton exchange membrane fuel cell (PEMFC), porous carbon nanofiber (PCNF), oxygen reduction reaction (ORR), electrocatalytic performance

## Abstract

Proton exchange membrane fuel cells (PEMFCs) represent an outstanding clean energy alternative for next-generation power sources. The PEMFC’s performance is mainly determined by the sluggish oxygen reduction reaction (ORR) that occurs in its cathode Therefore, the use of electrocatalysts with high electrocatalytic activity and stability for improving the ORR has been a vital direction for the commercialization of PEMFCs. In this article, porous carbon nanofibers (PCNFs) based on a polyacrylonitrile/polymethyl methacrylate (PAN/PMMA) precursor were fabricated by electrospinning followed by carbonization; then, the PCNFs were mixed together with carbon black (CB) in different mass ratios as a hybrid support for Pt nanoparticles. Pt nanoparticles were deposited on the hybrid support by the ethylene glycol reduction method, and the obtained series of Pt/(PCNF + CB) were used as the oxygen reduction electrocatalyst in the cathode. Their electrocatalytic properties, as well as those of Pt/C as a reference, were investigated by cyclic voltammetry scanning (CV) and linear sweep voltammetry (LSV). The results explained that Pt/(PCNF + CB) showed a higher electrochemical activity area and half-wave potential when the PCNF/CB mass ratio was 3/2 than that of commercial Pt/C. Furthermore, the half-wave potential of Pt/(PCNF + CB) only decreased by 4 mV, which was 86 mV lower than that of commercial Pt/C (90 mV) after 2000 ADT cycles, indicating that the incorporation of PCNFs to form a hybrid support could result in corrosion resistance.

## 1. Introduction

Since the environmental pollution caused by fossil energy restricts the sustainable development of society, exploring clean energy has become a necessity [1]. In particular, the proton exchange membrane fuel cell (PEMFC) is supposed to be the most promising sustainable power source for power generation due to its low operation temperature, high power density, and easy scalability [2,3]. Though many technical and associated fundamental breakthroughs have been achieved during the last couple of decades, there are still some problematic issues that need to be overcome before the extensive commercialization of PEMFCs [4]. The main reactions that occur in a PEMFC, including hydrogen oxidation reaction (HOR) in the anode and oxygen reduction reaction (ORR) in the cathode, are kinetically sluggish, especially ORR. Therefore, an electrocatalyst is essential for accelerating ORR and HOR [5].

Based on the excellent oxygen adsorption/dissociation activity of Pt, a Pt/C catalyst, that is, Pt nanoparticles (particle size ≈ 3–4 nm) supported on carbon black (CB), such as that supplied by Johnson Matthey Corp., E-TEK Corp., and TKK Corp., has become the most commonly used commercial ORR electrocatalyst so far [6]. Despite its favorable electrocatalytic activity, the high cost of precious Pt and the poor durability of its electrocatalytic activity are still issues. The main reason may be attributed to the fact that carbon black tends to be corroded in the harsh operating environment of PEMFCs, which might lead to the detachment and aggregation of Pt particles [7,8,9]. Therefore, many investigations have been conducted to improve the electrocatalytic activity and durability of the Pt catalyst through Pt morphological design with dimensional management of Pt particles [10,11] and Pt alloy exploration [12,13,14,15].

Changing the characteristics of the support is another strategy to raise the durability of Pt catalysts. When CB is replaced by carbon nanotubes [16,17], carbon nanofibers (CNFs) [18,19,20], or graphene [21], the durability of the Pt catalyst is extended. Wang et al. [22] fabricated PCNFs as a support on which to deposit Pt and found that the Pt/PCNF retained about 50% of its electrochemical activity area (ECSA) whereas Pt/C kept only 25% of its ECSA after 1000 cycles of CV sweeps, demonstrating the brilliant stability of PCNFs. Kim et al. [23] enhanced the metal–support interaction between Pt nanoclusters and the support by producing a three-dimensional (3D) graphene-like microporous carbon support. Their performance is affected significantly by factors such as the electronic connectivity between individual catalyst particles and the catalyst support, catalyst particle size, etc. [24]. These factors can be controlled by employing various modifications in the catalyst support to deposit the Pt particles uniformly and raise the interaction between the Pt particles and the support [25,26].

In order to improve the durability of the Pt catalyst while maintaining its performance, hybrid supports have been widely used to deposit Pt [27,28,29,30]. The 3D network formed by the hybrid support tends to provide a considerably high surface area and high internal connectivity [29,30]. Mu et al. [31] found that the 3D structure of the hybrid support consisting of graphene and CNF (RGO/CNF) offers more oxygen functional groups, superior stability, and lower charge transfer resistance than either RGO or CNF alone. Ruiz-Camacho et al. [32] prepared a hybrid carrier composed of GO and CB and found that Pt/GO-CB, when the GO/CB mass ratio was 1/1, showed better electrocatalytic activity as well as stability for ORR than any of the Pt/GO or Pt/CB catalysts due to the synergetic effect of GO with CB, which may be beneficial for modifying the Pt particle size, Pt dispersion, and Pt electronic properties. Kaplan et al. [33] prepared a Pt catalyst supported on the hybrid of CNF and CB for ORR and demonstrated that CB could act as a spacer in the hybrid support network to enhance the synergy between CB and CNF. Due to the interplay of electronic and lattice effects of the hybrid components, the electrocatalysts would make more active catalyst sites available for the PEMFC [34].

In this study, a hybrid support comprising PCNFs and CB was developed. The resulting porous structure could enhance the interaction between Pt NPs and the support so as to form a well-dispersed Pt network. Furthermore, the highly graphitized and fibrous PCNFs may preserve the 3D support network with their excellent corrosion resistance. By comparing the electrocatalytic activity of the prepared catalyst Pt/(PCNF + CB) with the commercial catalyst Pt/C (JM20), an optimal mass ratio of PCNF/CB of 3/2 was found to realize the superior electrocatalytic activity and durability of the Pt-based electrocatalyst.

## 2. Materials and Methods

### 2.1. Materials

Polyacrylonitrile (PAN, Mn = 150,000) and polymethyl methacrylate (PMMA, Mn = 20,000) were purchased from Sipin Technology Co., Ltd. (Hefei, China) as the carbon precursor and pore-forming agent, respectively. Carbon black (Vulcan XC-72) and Nafion (5 wt%) were supplied by Cabot Corp. and Dupont, respectively. The commercial catalyst JM20 (20 wt% Pt on Vulcan) was bought from Johnson Matthey Corp. as a reference sample. Dimethylformamide (DMF), hexachloroplatinic acid hexahydrate (H_2_PtCl_6_·6H_2_O), ethylene glycol (EG), sulfuric acid, and ethanol absolute were all obtained from Sinopharm Chemical Reagent Corp. (Shanghai, China). Deionized water and carbon paper (YLS-30T) were provided by the laboratory and Toray Corp. (Tokyo, Japan).

### 2.2. Fabrication of PCNF

PCNFs were fabricated by a common electrospinning method. PAN (1.5 g) and PMMA (0.64 g) were dissolved in DMF (10 g) solvent under stirring for 12 h to form a homogeneous solution. The resulting PAN/PMMA blend solution was loaded into a 10 mL syringe with a metal needle of inner diameter 0.33 mm. The spun blended nanofibers were collected on a grounded aluminum foil receiver horizontally, the distance between the needle and the aluminum foil was 18 cm. The temperature, humidity, extrusion rate, and high voltage were maintained at 25 °C, 40% R.H., 0.9 mL/h, 18 kV, respectively. The blended PAN/PMMA nanofiber mat was pre-oxidized in a muffle furnace at 250 °C for 1 h, and then carbonized at 1000 °C for 2 h under N_2_ atmosphere. The PMMA component was removed completely during carbonization, thus PCNFs were obtained.

### 2.3. Preparation of Electrocatalyst Pt/(PCNF + CB)

The hybrid of PCNFs and CB was used as the carrier to deposit Pt particle. To maintain the uniform dispersion of CB and PCNFs with each other, CB was firstly treated in HNO_3_ at 60 °C for 4 h and then separated from the solution and washed with a large amount of water. The dried CB was mixed with the PCNFs as a hybrid support with mass ratios of 20%, 30%, 40%, and 50%, respectively.

Pt/(PCNF + CB) electrocatalysts were fabricated by the ethylene glycol reduction method [35]: 30 mg of the hybrid support material with varied mass ratio of PCNFs and CB was added into 50 mL mixed solution (EG:DI water = 2:1) following ultrasonication for 1 h to get a homogeneous suspension. Then, a certain amount of H_2_PtCl_6_ solution (33.33 mg/mL) was dropped in the above suspension and ultrasonicated for 1 h. The resultant suspension was refluxed at 135 °C for 3 h with constant stirring under N_2_ atmosphere. Then, the suspension was purified by centrifugation with DI water and ethanol 3 times. The Pt/(PCNF + CB) catalyst powder was obtained after drying in a vacuum oven at 80 °C overnight. The Pt content was determined by inductively coupled plasma atomic emission spectroscopy (ICP-AES), and the results were 18.82 wt%, 18.66 wt%, 19.34 wt%, and 19.28 wt% for Pt/(PCNF + CB)—20%, Pt/(PCNF + CB)—30%, Pt/(PCNF + CB)—40%, and Pt/(PCNF + CB)—50%, respectively, and close to 20 wt% for commercial Pt/C.

### 2.4. Characterization of Samples

The morphology of PCNFs as well as that of the hybrid of PCNFs and CB was investigated by scanning electronic microscopy (SEM) with a field-emission scanning electron microscope (FESEM, Hitachi SU8010). The dispersion and particle size of Pt nanoparticles were examined by transmission electron microscopy (TEM, JEOL-2100F) operated at 200 kV. X-ray diffraction (XRD) patterns of Pt/(PCNF + CB) were acquired by a Bruker D2 Advance powder X-ray diffractometer with Cu Kα radiation at λ = 1.54 Å as the X-ray source.

### 2.5. Electrochemical Performance of Pt/(PCNF + CB)

The electrochemical performance of each Pt/(PCNF + CB) catalyst was measured by cyclic voltammetry (CV), linear sweep voltammetry (LSV), and accelerated durability test (ADT) on a three-electrode cell of an electrochemical workstation (CHI 760E Instruments). The ORR activity was examined with a rotation disk electrode (RDE) of a three-electrode cell, where Pt foil was used as the counter electrode and saturated calomel electrode as the reference electrode in O_2_-saturated 0.5 M H_2_SO_4_ solution. Specifically, the Pt/C and Pt/(PCNF + CB) catalyst suspensions were prepared by sonicating catalyst powder (2.0 mg), Nafion solution (50 μL), and ethanol (1 mL) for 15 min. Then, 20 μL of catalyst ink was dropped onto a pre-polished RDE (5 mm in diameter) and dried for the following measurements. LSV, CV, and ADT curves were obtained at a scanning rate of 5, 50, and 100 mV s^−1^, respectively, and the LSV electrode rotation speed was set as 1600 rpm. All potentials in the current manuscript were transformed to the reversible hydrogen electrode.

A Zn–air battery (ZAB) was used to investigate the electrocatalytic performance of Pt/(PCNF + CB) for ORR, thus only discharge behavior was measured. Except for the cathode, other parts including a Zn plate as the anode and the electrolyte (6 M KOH and 0.2 M Zn(CH_3_COO)_2_) remained as usual. The cathode was composed of the prepared catalyst loaded on carbon paper. The preparation of the cathode, in detail, included transferring the catalyst ink mentioned above, that is, Pt/(PCNF + CB)-40% and Pt/C as the ORR catalyst, respectively, into the airbrush, and spraying it on the carbon paper, which was then heat pressed at 60 °C. The ZAB test was performed at room temperature. The galvanostatic discharge curves were recorded by a Neware testing system at the current density of 10 mA cm^−2^.

## 3. Results and Discussion

### 3.1. Morphology of PCNF and Pt/(PCNF + CB)

The morphology of PCNFs is shown in Figure 1. Obviously, PCNFs exhibited a 1D structure and porous morphology with a diameter of around 500 nm, and the pores were distributed both on the surface and in the cross section. The presence of these pores could impart the roughness and activity of the fiber surface, which is beneficial for depositing and anchoring of Pt particles on the fiber surface.

The SEM images of the hybrid carriers with different CB mass ratios are shown in Figure 2a–d. It can be seen that the CB was dispersed in the PCNFs more homogeneously after pretreatment with HNO_3_. When CB was at a lower content of 20%, it was inclined to be concentrated in the cross section of the fiber due to its abundant mesoporous and defect structure, indicating good interaction between PCNFs and CB. As the content of CB increased to 30–40%, the PCNFs and CB dispersed with each other uniformly. That is, CB attached to the fiber surface and cross section more evenly. Meanwhile, the CB tended to serve as a link point to strengthen the connection between PCNFs. However, after the CB content reached 50%, the PCNFs were highly wrapped in CB, and the hybrid carrier was mainly composed of CB with high density.

In addition, the TEM images of the hybrid support are given in Figure 2e–h. The result also confirmed that PCNFs and CB in different weight ratios have excellent interaction with each other, which may establish a well-combined 3D-network support to provide a superior electron pathway.

The presence and distribution of Pt particles on the PCNF and CB hybrid support was observed by TEM as given in Figure 3a,b. Pt particles were loaded on the surface of the carbon support uniformly. Although the specific surface area of the PCNFs (150 m^2^/g) was lower than that of CB (245 m^2^/g) due to its large fiber size, the abundant porous structure of the PCNFs could increase defects, which could make the PCNF a more active surface for loading Pt and not only improve the dispersion of Pt nanoparticles but also enhance the binding interaction between Pt and the carrier. It revealed that the average size of Pt nanoparticles was in the range from 2.5 to 3.3 nm, which is similar to that of Pt/C (JM20 2.9 nm).

The XRD curves in Figure 3c reveal that all catalysts show a similar peak pattern. The diffraction peak at 2θ = 26° corresponds to the C (002) planes, which indicates that the graphite carbon support was fabricated successfully. Furthermore, the peaks at 2θ = 39°, 45°, 66°, and 82° correspond to the (111), (200), (220), and (311) crystal planes of Pt, respectively, which are consistent with the standard pattern of Pt [PDF#4-0802]. These mean that the Pt particles with a face-centered cubic structure were successfully deposited on the (PCNF + CB) hybrid carrier. Besides, the diffraction peak at 39° was the strongest among the crystal planes, suggesting that Pt had the best crystallinity along the (111) plane. According to the peak of Pt (220), the average size of Pt particles of each catalyst were calculated using the Scherrer formula to be in the range of 2–3 nm. This result is consistent with the TEM observations.

### 3.2. Electrocatalytic Activity of Pt/(PCNF + CB) toward ORR

Figure 4a represents the CVs of the Pt/(PCNF + CB) catalysts and JM20 as a reference in an oxygen-saturated 0.5 M H_2_SO_4_ solution at the scan rate of 50 mV s^−1^. All the samples showed similar hydrogen adsorption/desorption peaks at 0~0.4 V, and oxygen adsorption/desorption peaks from 0.6 V to 0.9 V, obviously. Since the electrocatalysis reaction mainly occurs at the three-phase interface of the electrode, the electrochemical activity area (ECSA) was used as a significant parameter to measure the catalytic activity of the catalysts. The equation is as follows: (1)$$\mathrm{ECSA}=\frac{Q}{0.21\;\times \;M}$$where Q (mC) is the electrical charge associated with monolayer adsorption of hydrogen on Pt with an assumed value of 210 μC/cm^2^, M (g) is the Pt loading on the working electrode. The value of ECSA was about 27, 54, 73, 66, and 72 m^2^/g_Pt_ for Pt/(PCNF + CB)—20%, Pt/(PCNF + CB)—30%, Pt/(PCNF + CB)—40%, Pt/(PCNF + CB)—50%, and Pt/C, respectively. Among those electrocatalysts, Pt/(PCNF + CB)—40% displayed the highest value of ECSA, which is very close to that of Pt/C. It confirmed that a proper mass ratio of PCNF to CB tends to form a superiorly dispersed carrier to decentralize Pt from agglomerating. Consequently, the exposure of Pt particles and the increment of the three-phase point in reaction could be the reason for better performance of the well-combined hybrid catalysts.

Figure 4b shows the LSV polarization curves at the electrode rotation speed of 1600 rpm and the scan rate of 5 mV s^−1^ in O_2_-saturated 0.5 M H_2_SO_4_. In sequence, each catalyst went through the kinetic control region, mixed dynamic region, and diffusion-limiting current region along with the negative scanning. The onset potential, half-wave potential, and limiting current density of the catalysts are listed in Table 1. Apparently, the onset potential of Pt/(PCNF + CB)—40% was 0.963 V, 19 mV higher than that of commercial Pt/C (0.944 V). In the meantime, the half-wave potential of Pt/(PCNF + CB)—40% reached 0.799 V, which is 25 mV higher than that of Pt/C (0.774 V), indicating that a hybrid support of CB with PCNFs in a proper ratio is superior compared with CB alone. The oxygen transport and oxygen reduction rate may be promoted by the present porous structure piled up with PCNFs and CB. Furthermore, the 3D network could provide a superior electron pathway to accelerate the electron transfer rate, which is helpful for increasing the limiting current density of the catalysts.

### 3.3. Durability of Pt/(PCNF + CB)

An accelerated durability test (ADT) of the electrocatalysts was performed by potential cycling the working electrode between 0 and 1.2 V (vs. RHE) for 2000 cycles with a scanning rate of 100 mV s^−1^ in O_2_-saturated 0.5 M H_2_SO_4_ solutions. Figure 5a,c record the cyclic voltammograms of Pt/C and Pt/(PCNF + CB)—40% after 100, 500, 1000, 1500, and 2000 cycles of aging. Obviously, as the number of cycles increased, the ECSA of the two catalysts decreased, indicating that both catalysts were attenuated unavoidably. The polarization curves before and after the ADT are recorded in Figure 5b,d. Notably, the half-wave potential of Pt/(PCNF + CB)—40% only decreased by 4 mV, whereas that of Pt/C showed a degradation of 90 mV after 2000 cycles, suggesting that Pt/(PCNF + CB)—40% has better stability than JM20.

The superior stability of Pt/(PCNF + CB)—40% could be explained by the diversity of Pt particles’ distribution after 2000 potential sweeps. As given in Figure 5e,f, Pt particles in Pt/C were agglomerated and dissolved seriously after 2000 cycles of potential sweeps, whereas Pt particles in Pt/(PCNF + CB)—40% still maintained a relatively uniform dispersion state on the PCNF surface, but the accumulation of Pt particles in CB occurred, suggesting more interaction between Pt particles and PCNFs due to the porous surface. In this way, PCNFs possess the ability to extend the durability of the electrocatalyst by maintaining the uniform Pt distribution on PCNFs and making full use of electrocatalytic area of Pt particles.

### 3.4. Discharge Behavior of Zinc–Air Battery with Pt/(PCNF + CB) as ORR Catalyst

In order to investigate the discharge performance of the electrocatalysts more practically, Pt/(PCNF + CB)—40% was applied as the cathode catalyst of a zinc–air battery to initiate the oxygen reduction reaction, and commercial Pt/C was used as a reference catalyst. As shown in Figure 6, both the assembled ZABs had a similar discharge voltage at 10 mA cm^−2^. Moreover, the discharge duration time of the Pt/(PCNF + CB)—40% catalyst reached 80 h, which was 5 h longer than that of Pt/C (75 h) under the same conditions. This outcome is consistent with the accelerated durability test result mentioned above.

## 4. Conclusions

In summary, Pt/(PCNF + CB) electrocatalysts were fabricated successfully by the ethylene glycol reduction method based on the hybrid support of PCNFs and CB. The hybrid network could prevent both PCNFs and CB from agglomerating, PCNF/CB with a 3/2 mass ratio may constitute a superior 3D porous electron network to provide more accessible catalytic sites to improve the electron transfer rate and electrocatalytic activity. Thus, the Pt/(PCNF + CB)-40% catalyst showed higher electrocatalytic activity with higher half-wave potential than the commercial Pt/C catalyst. Furthermore, the Pt/(PCNF + CB)-40% catalyst revealed better catalytic durability than Pt/C after 2000 ADT cycles; this may be attributed to greater interaction between Pt particles and the PCNFs as well as the high graphitization of PCNFs, which is conducive to enhancing the resistance of the support to corrosion.

## Figures and Tables

**Figure 1 materials-15-04560-f001:**
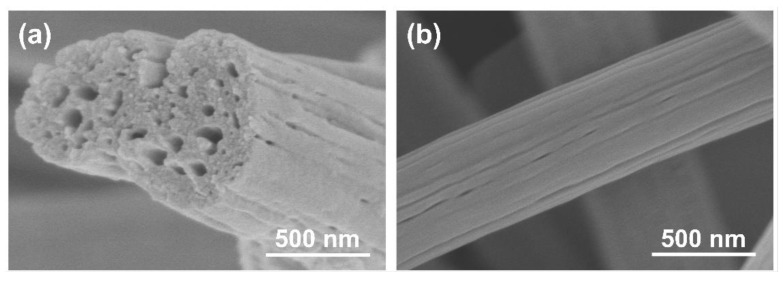
SEM images of PCNF: (**a**) cross section; (**b**) surface.

**Figure 2 materials-15-04560-f002:**
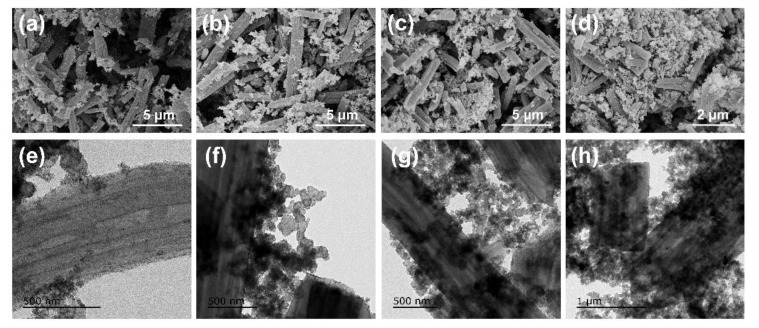
SEM images of electrocatalysts: (**a**) Pt/(PCNF + CB)—20%; (**b**) Pt/(PCNF + CB)—30%; (**c**) Pt/(PCNF + CB)—40%; (**d**) Pt/(PCNF + CB)—50%. TEM images of electrocatalysts: (**e**) Pt/(PCNF + CB)—20%; (**f**) Pt/(PCNF + CB)—30%; (**g**) Pt/(PCNF + CB)—40%; (**h**) Pt/(PCNF + CB)—50%.

**Figure 3 materials-15-04560-f003:**
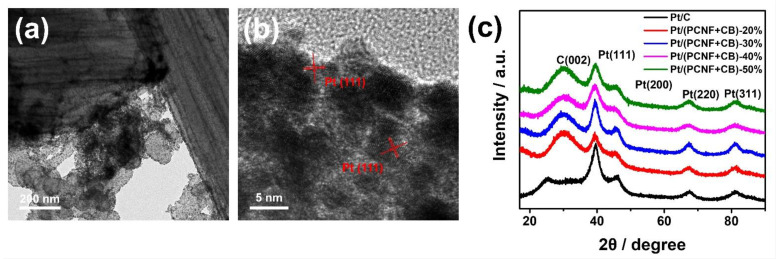
TEM images (**a**,**b**) and XRD patterns (**c**) of electrocatalysts.

**Figure 4 materials-15-04560-f004:**
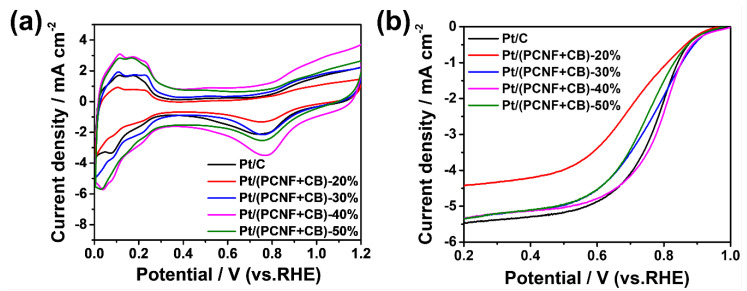
CV and LSV curves of different electrocatalysts: (**a**) in 0.5 M H_2_SO_4_ under the scanning rate of 50 mV s^−1^; (**b**) under the scanning rate of 5 mV s^−1^ and 1600 rpm electrode rotation speed.

**Figure 5 materials-15-04560-f005:**
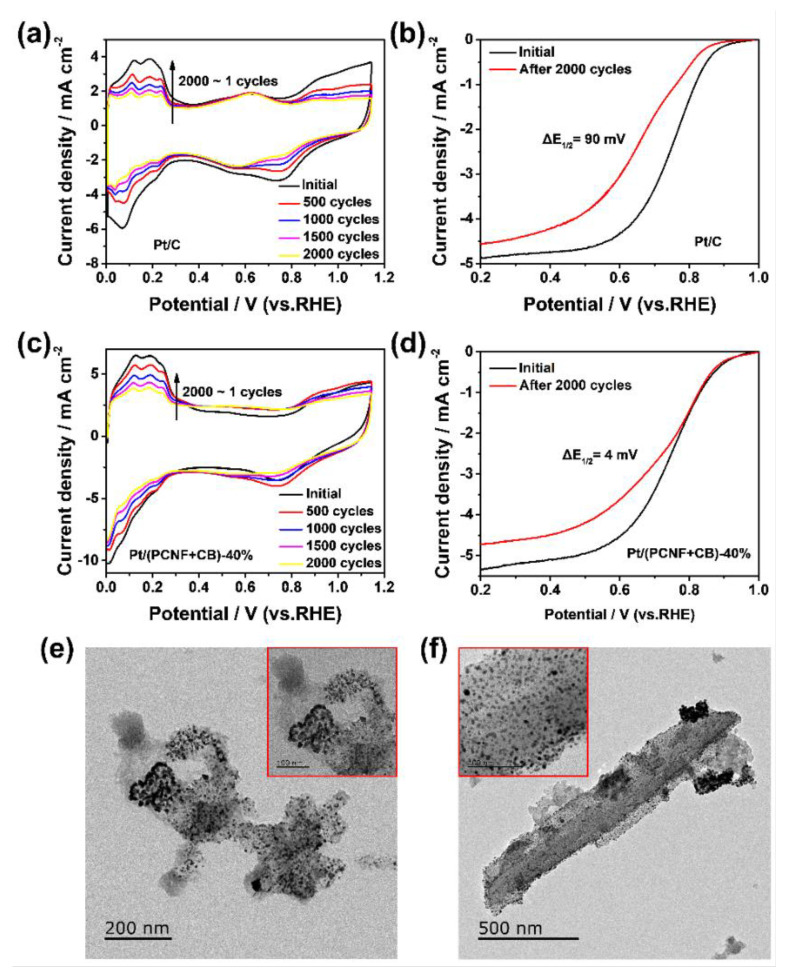
Cyclic voltammetry curves of (**a**) commercial Pt/C and (**c**) Pt/(PCNF + CB)—40%; ORR polarization curves of (**b**) Pt/C and (**d**) Pt/(PCNF + CB)—40% before and after 2000 potential sweeps; TEM images of electrocatalysts (**e**) Pt/C and (**f**) Pt/(PCNF + CB) after 2000 potential sweeps.

**Figure 6 materials-15-04560-f006:**
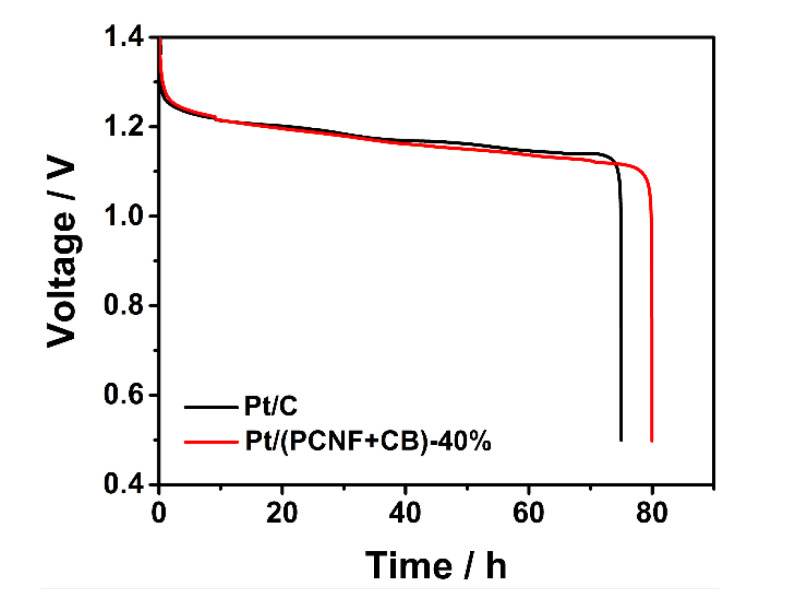
Galvanostatic discharge curves at 10 mA cm^−2^ of the indicated catalysts.

**Table 1 materials-15-04560-t001:** Total results of the CV and LSV measurements.

Electrocatalysts	ECSA (m^2^/g_Pt_)	Half-Wave Potential (V)	Onset Potential (V)	Limiting Current Density (mA cm^−2^)
Pt/(PCNF + CB)-20%	27	0.703	0.928	4.43
Pt/(PCNF + CB)-30%	54	0.756	0.945	5.34
Pt/(PCNF + CB)-40%	73	0.799	0.963	5.34
Pt/(PCNF + CB)-50%	66	0.740	0.942	5.36
Pt/C	72	0.774	0.944	5.48

## Data Availability

Not applicable.

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
