# Peer review of "The Oxygen Reduction Performance of Pt Supported on the Hybrid of Porous Carbon Nanofibers and Carbon Black"

_materials, 2022, doi:10.3390/ma15134560_

Round 1

Reviewer 1 Report

The document is relevant for new energy applications, however, it is necessary to clarify some points in the manuscript.

1. Section 2.2. Why were the conditions chosen for the synthesis: PAN (1.5 g) and PMMA (0.64 g) were dissolved in DMF (10 g) solvent. Based on what criteria were these conditions established?

2. What are you looking for with the pre-oxidized at 250 °C and what will be the structures formed at this temperature? Was oxygen fed to the muffle chamber or was it developed under static oxygen conditions? Explain.

3. Line 157 it is mentioned: It can be seen that the CB was dispersed in PCNF more homogeneously after pre-treating by HNO3, but why does it allude to the fact that pre-treatment is better if no pre-treatment micrographs were presented? Could you clarify this idea or attach the corresponding micrographs to evidence what you say.

4. Line 170 it is mentioned that the abundant porous structure of PCNF can not only increase the specific surface area. Was measurement the specific surface area of ​​solids?  If so, attach the results obtained for surface area and porosity.

5.  In Figure 3 a-b, Does the TEM micrograph has only Pt/ PCNF without CB? The behavior will be similar for the Pt/PCNF-CB-40 %?

6. Line 178: it is mentioned that the peaks at 2θ = 39°, 45°, 66° and 82° correspond to the (111), (200), (220), and (311) crystal planes of Pt, respectively, but does the structure of platinum correspond to metallic oxide or metallic platinum? Can you explain to which structure it belongs and what benefits would it have in the reaction?

Reviewer 2 Report

This manuscript reported the preparation and characterization of Pt/(PCNF+CB) electrocatalyst for ORR and potential application in PEMFC. The paper could be published after addressing the following points.

1.      Abstract: Abbreviations should be defined after the first mention in the manuscript, e.g. PAN/PMMA.

2.      Experimental section: In section 2.2, it should be commented that PCNF were obtained by electrospinning in the first sentence.

3.      The Pt loading of the synthesized catalyst powders and commercial Pt/C s is no mentioned in the experimental section.

4.      The authors deal that PCNF exhibits pores in both the surface a cross-section, which are beneficial for depositing the Pt particles; however, the inner pores in the cross-section is nor relevant because Pt is deposited on the PCNF surface. Have the authors determined the BET surface?

5.      It is recommendable to include SEM image at higher magnification to visualize the good interaction between PCNF and CB. This should be also confirmed by TEM image.

6.      The XRD peak at 2Theta=20º is not commented.

7.      The equation of ECSA in page 4, line 192 should be given in a separate file in the document.

8.      The degradation of commercial Pt/C during the stability test was associated with agglomeration of Pt particles (Fig. 5e); however, it is impossible to visualize particles of 2-3 nm at a magnification of 500 nm.

9.      The list of references need to be improved with recent and relevant works in this field. Only 6 references since 2019 are included.

Reviewer 3 Report

Title: The Oxygen Reduction Performance of Pt Supported on the Hybrid of Porous Carbon Nanofibers and Carbon Black

Authors: Tongyu Zhou, Jingjing Zhang, Shenglin Yang, Junhong Jin, Biao Wang, Guang Li

General Comments:

o  In this paper, series of Pt/(PCNF+CB) were used as electrocatalyst for the oxygen reduction. The catalytic activity is analyzed and discussed.

o   The structure of the article fulfills the structure of a research article.

o   Four keywords are included by the authors.

  • The Introduction section provide sufficient background information for readers in the immediate field to understand the problem that this study addresses.

o   The authors present in the Materials and Methods section shortly the reagents, the experimental method and the equipment used.

o   In the Results section, the authors present and interpret the results of the performed experiments.

o   The paper ends with the Conclusions part. In this section the authors mention the conclusions of their research study.

I suggest to Reconsider after Minor Revisions for the following reasons:

1. the authors should highlight the novelty of the present work;

2. page 3, line136: please note the scanning rate must in ascending order: 5, 50 and 100 mV/s;

Round 2

Reviewer 1 Report

The authors realized the corrections and added information to the document,
but it is necessary to add the reference (https://doi.org/10.1016/j.electacta.2015.03.142) in section 2.2. manufacture of PCNF (response to point 1) for clarity.

Reviewer 2 Report

The manuscript has been improved and can be accepted.

This manuscript is a resubmission of an earlier submission. The following is a list of the peer review reports and author responses from that submission.